# Aspects of Psychiatric Comorbidities in Breast Cancer Patients in Tertiary Hospitals Due to COVID-19 Outbreak in South Korea: A Single Center Longitudinal Cohort Study

**DOI:** 10.3390/medicina58050560

**Published:** 2022-04-19

**Authors:** Jeongmin Park, Seonhwa Kim, Jaesung Heo

**Affiliations:** 1Department of Convergence Healthcare Medicine, Ajou University School of Medicine, Suwon 16499, Korea; pjmpsj12@gmail.com (J.P.); sh092433@naver.com (S.K.); 2Department of Radiation Oncology, Ajou University School of Medicine, 164 Worldcup-ro, Yeongtong-gu, Suwon 16499, Korea

**Keywords:** breast cancer, mental disorders, psychiatry, COVID-19, pandemic

## Abstract

*Background and Objectives*: This study aimed to analyze the prevalence of mental disorders in patients with breast cancer at Ajou University Hospital. In addition, the patterns and prevalence of mental disorders according to the occurrence of coronavirus disease (COVID-19) were analyzed. *Materials and Methods*: From 1 January 2008 to 30 June 2021, psychiatric disorders were identified in 5174 female patients diagnosed with breast cancer at Ajou University Hospital. Based on the time when COVID-19 occurred, the pattern of onset of mental disorders in patients with breast cancer was analyzed. In addition, the prevalence of mental disorders according to the time of breast cancer diagnosis and age was evaluated. *Results*: A year before the diagnosis of breast cancer, 371 patients were diagnosed with a mental disorder. Of these, 201 patients were diagnosed with stress and adjustment disorders, and 97 patients had anxiety disorders. The overall frequency of psychiatric disorders after breast cancer diagnosis peaked two months later. Among psychiatric disorders reported before the COVID-19 pandemic, the proportion of stress/adaptation disorders was 52%, and among psychiatric disorders reported after the pandemic, it was significantly higher at 94.7%. Anxiety was found to be high in the elderly group aged ≥ 60 years, and the prevalence of stress and adjustment disorders tended to increase in the non-elderly group. *Conclusions*: Breast cancer patients showed different patterns of psychiatric disorders according to age, time of breast cancer diagnosis, and the occurrence of COVID-19. Owing to the COVID-19 pandemic, delays in treatment and anxiety about infection have increased the rate of stress and adjustment disorders in cancer patients. Mental health management during the pandemic and after cancer diagnosis can improve the quality of life of patients with cancer.

## 1. Introduction

The Coronavirus Infectious Disease-19 (COVID-19) pandemic, as we know it, is still disrupting the world and having a major impact on healthcare systems around the world [1]. In Korea, the use of medical services has decreased since the COVID-19 pandemic [2]. According to the main statistics of medical expenses in the first half of 2020, the number of visits decreased by 12.5%, and the number of days of outpatient visits decreased by 15.2% compared with the first half of 2019 [3].

In the first half of 2020, 15.6% of medical service users visited medical institutions and felt anxious about contracting an infection [2]. This is more than double the figure in the first half of 2019 (6.2%). Consequently, concerns about SARS-CoV-2 infection have reduced the use of medical services. COVID-19 has also affected patients with severe diseases including cancer [4]. Treatment protocols for cancer patients have changed, and healthcare provisions have decreased [4,5]. The delay in the diagnosis and treatment of the disease has caused psychological pain among cancer patients [6].

Breast cancer is the most common cancer in women and has the highest incidence among all types of cancers worldwide [7]. Breast cancer patients experience more psychological depression than general cancer patients because of menopause and hormone reduction after treatment [8]. In addition, breast cancer patients experience symptoms such as fatigue, sleep disturbance, depression, and anxiety during the course of treatment because of various adjuvant therapies in addition to surgery [9]. Psychological problems that accompany cancer patients reduce adherence to treatment and survival rates [10]; therefore, it is necessary to take measures to deal with the psychological problems of cancer patients and the changes in the medical environment brought by the pandemic. Appropriate clinical management through identification of the onset pattern of the psychiatric disorder can improve the quality of life of cancer patients.

The purpose of this study was to investigate the impact of the COVID-19 outbreak on mental disorders of breast cancer patients in tertiary hospitals. In addition, this study aims to analyze the overall pattern of mental disorders from 1 year before breast cancer diagnosis to after treatment.

## 2. Materials and Methods

For this retrospective, in a single-center study, patients’ data were collected from 1 January 2008 to 30 June 2021, at the Ajou University Hospital. The library had data on 5199 patients diagnosed with breast cancer who visited Ajou University Hospital during this period. Among them, 5174 patients were classified as the final study subjects, excluding 8 patients with missing values and 17 males. The subjects of the study were patients who received C50 and D05 as the main diagnosis based on the Korean Standard Classification of Diseases and Causes, 8th edition (KCD-8), which was revised based on the 10th International Classification of Diseases (ICD-10). The breast cancer library used in the study includes demographic information and clinical information, such as the date of breast cancer diagnosis, mental disorder medical records, and disease codes. This was de-identified data, and the study subject’s consent was waived. The study was approved by the Institutional Review Board (IRB) of Ajou University Hospital.

To focus on the impact of newly occurring mental disorders during the cancer diagnosis process, we excluded individuals with previous psychiatric disorders over 1 year before the date of the diagnosis of breast cancer. In addition, we limited the study to within a period of 6 months after the diagnosis of cancer. The first diagnosis of psychiatric disorders was identified using data from the first psychiatric visits of inpatients and outpatients at Ajou University Hospital. Mental disorders were also classified into 5 groups according to the disease code based on the KCD-8. Patients diagnosed with F32 and F33 were defined as having depression, F40 and F41 as anxiety, F44 and F45 as body type/conversion disorder, F43 as stress response/control disorder, and F10–F19 as substance abuse disorder.

We conducted a prevalence analysis on the first diagnosis of mental disorders from 1 year before the diagnosis of breast cancer to at least 6 months after diagnosis. The mental disorder patterns of patients with breast cancer before and after the onset of the COVID-19 pandemic were analyzed. The prevalence patterns of the group before and after the diagnosis of breast cancer and the characteristics of mental disorders by age group were also evaluated. Descriptive statistics were calculated to estimate the percentage of mental disorders and to estimate age-specific characteristics of breast cancer patients. All statistical analyses were performed using the R 4.1.1.

## 3. Results

From 1 January 2008 to 30 June 2021, a total of 5174 female patients were diagnosed with breast cancer at Ajou University Hospital. The median age at diagnosis for breast cancer patients was 49 years, and for breast cancer patients with mental disorders, 48 years. A total of 371 cancer patients, all female, visited the hospital due to mental disorders one year before they were diagnosed with cancer. Table 1 shows the frequency of mental disorders in breast cancer patients. As for the prevalence of mental disorders in breast cancer patients, stress/adaptation disorder was the most common with 201 (54.2%), followed by depression with 97 (26.1%), and anxiety with 60 (16.2%).

Based on the date of the diagnosis of breast cancer, the proportion of patients with mental disorders in the group before and after the onset of the COVID-19 pandemic was calculated. Among all patients with breast cancer, 4498 were in the pre-pandemic group and 676 were in the post-pandemic group. Before the outbreak of the pandemic, the number of patients with mental disorders was 352 out of 4498 (7.8%). After the pandemic, 19 out of 676 (2.8%) had mental disorders. The frequency of mental disorders before and after COVID-19 is shown in Table 2. Mental disorders reported prior to the COVID-19 pandemic were stress/adaptation disorders in 183 cases (52%), followed by depression in 97 cases (27.6%), and anxiety in 60 cases (17%). As for the mental disorders reported after COVID-19, stress/adaptation disorders accounted for the most with 18 cases (94.7%), and the rate was higher than before the outbreak of COVID-19. The ratio of the stress adaptation disorder based on total breast cancer patients was found to be 4.06% (183 out of 4498) before COVID-19 and 2.7% (18 out of 676) after COVID-19. Other mental disorder groups also showed lower prevalence rates after the COVID-19 pandemic.

The incidence ratios of mental disorders by age group are shown in Figure 1. Depression and anxiety were relatively high in elderly breast cancer patients, and stress/adaptation disorders were high in non-elderly people under 60 years of age.

The frequency density by disease also peaked within 2 months after diagnosis. There was a difference in the peak times of the mental disorders. The overall frequency density of mental disorders increased from 12 months before the diagnosis of breast cancer to 2 months after diagnosis (Figure 2). The frequency density changed rapidly before and after cancer diagnosis. The increase in frequency density for mental disorders was higher in patients aged around 60 years than in the other age groups.

The frequency density by disease also peaked within 2 months after diagnosis (Figure 3). There was a difference in the peak times of the mental disorders. Depression and anxiety frequency peaked at the time of breast cancer diagnosis, whereas stress peaked at 2 months after diagnosis of cancer. The condition with the highest rate of increase after cancer diagnosis was found to be stress/adaptation disorders.

## 4. Discussion

Based on 20 January 2020, when the first COVID-19 patient occurred in Korea, the prevalence of mental disorders in breast cancer patients before and after the onset of COVID-19 was analyzed. Upon examining the proportion of mental disorders in the pre-outbreak and post-outbreak groups among all breast cancer patients, the pre-outbreak group showed a higher prevalence of mental disorders. In a previous study, after the outbreak of COVID-19, the outpatient treatment rate of psychiatric outpatients at tertiary hospitals in Korea decreased by approximately 13.1%, and in the UK, the number of cases of mental disorder diagnosed decreased by 50% compared to the previous year [11,12]. During the COVID-19 pandemic, outpatients with mental disorders had poorer access to appropriate psychiatric care [11]. A pandemic can increase the risk of psychiatric problems and worsen health conditions [13]. Cancer patients experience fear, anxiety, and depression for reasons such as delay or discontinuation of treatment during a pandemic [14]. Cancer patients experience psychological pain due to the COVID-19 outbreak, but it seems that there are limitations in the diagnostic aspect of mental disorders due to the fear of infection and changes in the medical environment during the pandemic. As such, the COVID-19 pandemic may become an obstacle to early detection and diagnosis of mental disorders and may further aggravate the patient’s situation in the future; therefore, the medical staff should pay greater attention to the psychological aspect of the patient.

As a result of comparing the prevalence of mental illness in the groups before and after the outbreak of COVID-19, stress/adaptation disorders accounted for 52% of the total 352 mental disorders before the outbreak of COVID-19, with 183 cases. After COVID-19, stress/adaptation disorders accounted for 94.7% of the 19 cases of mental disorders. These results suggest that the pandemic can cause stress/adaptation disorder symptoms in patients with cancer. Furthermore, a comparison of the stress related to COVID-19 between the general public and cancer patients revealed that the prevalence of stress in cancer patients was higher [15]. COVID-19 has resulted in delays and changes in the treatment protocols for patients with cancer [4], which can cause psychosocial distress [6]. Cancer patients are susceptible to respiratory virus infection due to reduced immunity, and the risk of SARS-CoV-2 infection is higher than that in the general population [16]. Anxiety over COVID-19 infection in cancer patients can affect their mental health [17]. The results of this study suggest the need for coping measures to minimize stress symptoms and psychological distress in cancer patients during social isolation due to infectious diseases. Further research should be carried out to identify methods to reduce the psychological pain of cancer patients during the pandemic period, prevent the contraction of infectious diseases, and cope with delays in treatment protocols.

Breast cancer patients experience a high level of stress during the period from clinical suspicion before diagnosis to final cancer diagnosis [18]. Cancer patients have a sharp increase in the risk of mental disorders from one year before they are diagnosed with cancer [19]. In addition, after the initial diagnosis was made, the patient had to go through several stages of examinations and laboratory tests; therefore, the frequency and pattern of mental disorders from one year before the diagnosis of breast cancer to the period after diagnosis were analyzed. As a result of classifying the mental disorder patient group based on the time of diagnosis of breast cancer, there were six patients in the pre-diagnosis group and 365 patients in the post-diagnosis group. The rates of depression and anxiety were highest in the pre-diagnosis group, but they decreased after diagnosis. In the period after breast cancer diagnosis, the frequency of patients diagnosed with stress/adaptation disorders increased rapidly (Figure 3). In a previous study, analysis of mental disorder patterns based on Health Insurance Review & Assessment Service (HIRA) national breast cancer patient data, showed that stress/adaptation disorders had the largest increase in trend [20]. The results of the research conducted based on public data and the results of this study were consistent with data from a single medical institution. These findings suggest that is essential to manage stress and adaptation disorders in cancer patients immediately after cancer diagnosis. In the case of disease incidence by age, there was a difference in the pattern of mental disorders between the elderly and non-elderly, with a cut-off age of 60 years (Figure 1). The provision of psychological management requires a variety of approaches depending on the age group-specific mental disorder symptoms of cancer patients.

Among all breast cancer patients at Ajou University Hospital, the proportion of patients diagnosed with a mental disorder was 7.17% (*n* = 371). The average of the proportion of mental disorder patients among the total female population in Korea calculated from the statistics of the Health Insurance Review and Assessment Service and the census data from the National Statistical Office from 2010 to 2020 is 3.94% [21,22]. These results confirm that breast cancer patients are more vulnerable to mental disorders than the general population. In a previous study, when breast cancer patients received psychological services in addition to medical treatment, the risk of cancer recurrence and death was reduced [23]. In the United States, a patient navigator program is provided for assistance and guidance in accessing cancer treatment systems [24]. This is a system developed to close the gap in access to medical care, and the navigator provides psychological, social, and physical support during the clinical treatment process of patients and affects the quality of life of patients [25,26]. The results of a navigation service satisfaction survey on breast cancer patients showed high satisfaction with psychosocial support [27]. In Korea, a project for cancer survivors who have completed active treatment is being implemented, but no consideration has been given to patients undergoing treatment in actual clinical practice [28]. In breast cancer patients, mental disorders showed the highest incidence immediately after cancer diagnosis (Figure 2 and Figure 3). During this period, the patients stayed at the medical institution for diagnosis and treatment; therefore, various efforts are needed, such as the placement of psycho-oncology experts in medical institutions and the provision of cancer patient emotional management education for medical staff.

This study had some limitations. First, the measurement of psychiatric disorders was assigned based on the diagnosed disease code and was not formalized in a research setting. Each medical institution has different evaluation tools for mental disorders, and tertiary hospitals have the lowest rate of use of evaluation tools at the initial diagnosis of depression [29]. Depression is diagnosed based on the medical experience of doctors; hence, the rate of use of the evaluation tool is low. As such, there is a possibility that the subjective intervention of medical staff may be reflected in the diagnosis of mental disorders. The shortcomings of the disease code can be addressed by conducting structured clinical interviews with breast cancer patients or by considering clinical diagnosis, drug use, and timing of treatment. Second, since this study analyzed the prevalence patterns by focusing on the first diagnosis of mental disorders and the time of diagnosis of breast cancer, time series analysis of mental disorders and psychosocial factors were not considered. The survival rate of breast cancer is high, but there are long-term and short-term side effects due to various high-dose therapeutic agents [30]. In addition, breast cancer patients experience more symptoms such as depression, anxiety, and sleep disturbance during chemotherapy than patients with other forms of cancer [9]. Further studies on the psychiatric comorbidities occurring during the adjuvant therapy period and the follow-up period and the psychosocial factors that cause mental disorders should be conducted. Third, as a result of confirming the pattern of mental illness before and after breast cancer diagnosis in this study, there was a significant difference in the number of patients in each group. Mental disorders are most frequently diagnosed in primary medical institutions [31]. This study uses only the claims data of Ajou University Hospital; therefore, there is no medical record of mental disorders from an external medical institution. Thus, there was a limit to the analysis of the prevalence of mental disorders before and after cancer diagnosis; therefore, research including psychiatric diagnosis data before breast cancer diagnosis in breast cancer patients is needed.

## 5. Conclusions

The prevalence of psychiatric disorders in breast cancer patients differed according to age, time of breast cancer diagnosis, and the outbreak of the COVID-19. The prevalence of mental disorders in patients with breast cancer increased sharply right after cancer diagnosis; therefore, the management of mental disorders immediately after diagnosis can improve the quality of life of patients with cancer. COVID-19 has a psychological impact on cancer patients due to delayed treatment and anxiety about infection. Further studies on coping measures and mental management should be conducted in consideration of the treatment environment of cancer patients during the pandemic.

## Figures and Tables

**Figure 1 medicina-58-00560-f001:**
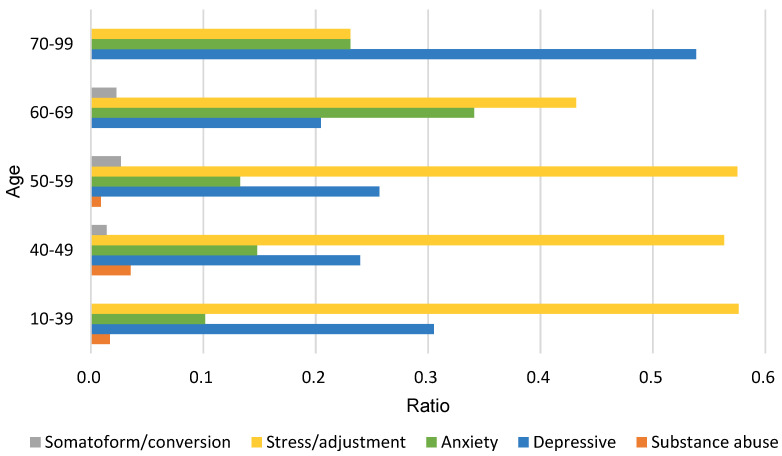
The distribution of mental disorders by age group.

**Figure 2 medicina-58-00560-f002:**
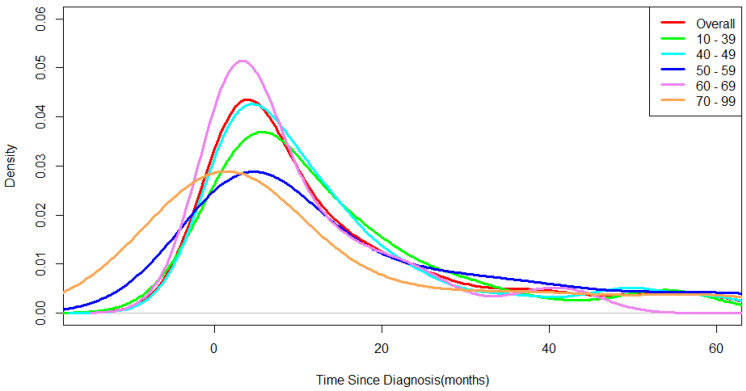
The frequency density of mental disorders, stratified by age group in breast cancer patient.

**Figure 3 medicina-58-00560-f003:**
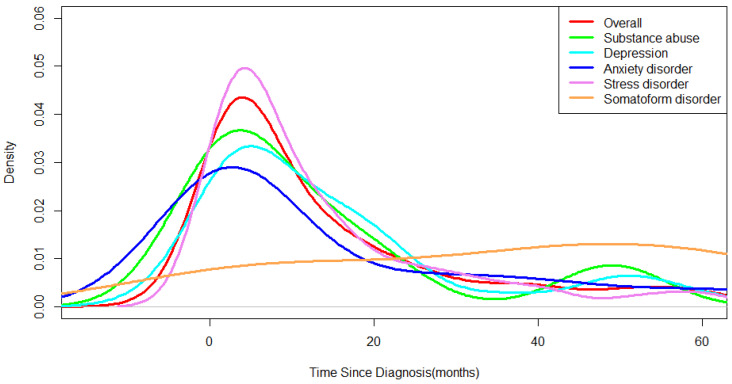
The frequency density by metal disorder in breast cancer patient.

**Table 1 medicina-58-00560-t001:** The frequency of mental disorders in breast cancer patients (*n* = 5174).

Age	Total Number of Breast Cancer	Mental Disorder	Substance Abuse	Depressive Disorder	Anxiety Disorder	Stress/Adjustment Disorder	Somatoform /Conversion Disorder
10–39	643	59	1	18	6	34	0
40–49	2054	142	5	34	21	80	2
50–59	1474	113	1	29	15	65	3
60–69	672	44	0	9	15	19	1
70–99	331	13	0	7	3	3	0
Total	5174	371	7	97	60	201	6

**Table 2 medicina-58-00560-t002:** Mental disorder before COVID-19 and post COVID-19 (*n* = 371).

Patient Characteristics	No. of Patients (%)					
	Mental Disorder	Substance Abuse	Depressive Disorder	Anxiety Disorder	Stress/Adjustment Disorder	Somatoform/Conversion Disorder
before COVID-19	352	6 (1.7)	97 (27.5)	60 (17.0)	183 (51.9)	6 (1.7)
after COVID-19	19	1 (5.2)	-	-	18 (94.8)	-

## Data Availability

The data used in the study are potentially identifiable and are not publicly available. The raw claims datasets generated and/or analyzed during the study are not publicly available due ethical restrictions by the institution.

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
