# Peer review of "Aspects of Psychiatric Comorbidities in Breast Cancer Patients in Tertiary Hospitals Due to COVID-19 Outbreak in South Korea: A Single Center Longitudinal Cohort Study"

_medicina, 2022, doi:10.3390/medicina58050560_

Round 1
Reviewer 1 Report
Title: Analysis of the prevalence of mental disorders in breast cancer patients before and after the outbreak of COVID-19
The authors report analysis of the prevalence of mental disorders in breast cancer patients before and after the outbreak of COVID-19. Although it shows that COVID‐19 has an important impact on the prevalence of mental disorders in breast cancer patients, the manuscript does not accurately describe the real situation after the outbreak of COVID-19, owing to too few data after 2020. Moreover, I have a few minor and major points which need to be addressed.
Minor points:
- The first sentence in Introduction should being replaced with “The Coronavirus disease (COVID‐19) pandemic is still disrupting the world as we knew it, and has exerted a major impact on healthcare systems worldwide [1]”
- The lower resolution limits the quantity of Figure1 and Figure2.
- The labels of coordinates should be exhibited in Figure1 and Figure2 as shown in Figure 3.
- Line 182: There is “(Figures 3 and 5)” in the manuscript, but I don't see Figure 5 in the paper.
Major points:
- The quality control process of data using 95%confidence interval (CI) method [11] (Williams, R. et al. 2020) should be showed in supplementary Information added to the end of this manuscript. [11. Williams, R.; Jenkins, D.A.; Ashcroft, D.M.; Brown, B.; Campbell, S.; Carr, M.J.; Cheraghi‐Sohi, S.; Kapur, N.; Thomas, O.; Webb, R.T. Diagnosis of physical and mental health conditions in primary care during the COVID‐19 pandemic: a retrospective cohort study. The Lancet Public Health 2020, 5, e543‐e550, doi: https://doi.org/10.1016/S2468‐2667(20)30201‐2]
- Lines 110-112: Authors only calculated the proportion of the stress/ adaptation disorder in the total number of mental disorders of breast cancer patients. Should authors display the ratio of stress/adaptation disorder to total breast cancer patients? (i.e., the numbers of patients with stress/adjustment disorder were 183 out of 4515 (4.1%) and 18 out of 676 (2.7%) stress/adjustment disorder respectively before and after the outbreak of the pandemic, and this result is consistent with the main conclusion of the paper).
- Lines 120-122: The pre‐diagnosis group only has 6 patients, and the number of pre‐diagnosis patients is too low.
- The authors should have more details of the patients, especially the clinical manifestations of the index case and the standard clinical diagnostic criteria used for mental disorders diagnosis in Materials and Methods of the manuscript.
- Lines 193-195: The authors should refer to the mean value of incidence of mental disorders among the general population from 2008 to 2021 (or 2020), not the value of prevalence of mental disorders in 2020 (Refs. 19 in line 318). [19. National Mental Health Statistics 2020; Ministry of Health and Welfare: National Center for Mental Health, 2020.]
Reviewer 2 Report
Dear authors, thank you for the interesting study.
Although the idea is good, there are some concerns to be addressed:
- The title of the manuscript. I can suggest to use the CONSORT rules for the manuscripts, describing the clinical trials (http://www.consort-statement.org/).
- Title should be SPICED, that is, it should include Setting, Population, Intervention, Condition, End-point, and Design
- Materials and Methods. Please indicate the type of the study (A Single-Center Prospective Cohort Study), randomization procedures, sample size calculations.
- Lines 59-61.” In this study, the impact of COVID‐19 on mental disorders in breast cancer patients was evaluated”. Please write the clear aim of the study and the null hypothesis.
- Lines 92. “Of these, 5,174 were women and 17 were men.” Breast cancer is about 100 times less common among white men than among white women (https://www.cancer.org/cancer/breast-cancer-in-men/about/key-statistics.html) thus it is better to exclude the men from the study. As the pathophysiological mechanisms of the depression development may be different due to the influence of sex hormones and different treatment strategies used.
- Line 107. Table 2. please indicate the frequency in %. Please add and discuss the data on general population as well.
- Line 120-121. Probably it is not correct to compare the groups of six and 365 patients. It is better to delete that part (and simply mention the tendency in the Discussion).
- Line 136 and the Discussion. “The frequency density by disease also peaked within 2 months after diagnosis. There was a difference in the peak times of the mental disorders.” please explain how many times each patient was checked for the Mental disorders (MD). Otherwise, the revealed peaks could be explained by the time-course of treatment (for example, after the initial diagnosis was made, the patient had to go through several stages of examinations and laboratory tests, and the test for the MD was among them).
- Lines 157-161. Why are you discussing only the ratio and not the change (reduction) in the frequency?
- Please revise and make it clear, and focus on the aims of the manuscript.
Round 2
Reviewer 1 Report
Title: Analysis of the prevalence of mental disorders in breast cancer patients before and after the outbreak of COVID-19
I agree to publish the revised paper (medicina-1661696) in Medicina.
Author Response
I agree to publish the revised paper (medicina-1661696) in Medicina.
=> Response : We appreciate your favorable comments for our manuscript.
Reviewer 2 Report
Dear authors, according to the text, "For this retrospective, single-center study, patients data were collected from January 1, 2008 to June 30, 2021, at the Ajou University Hospital" and thus it is not correct to title the manuscript as the "Aspects of psychiatric comorbidities in breast cancer patients in tertiary hospitals due to COVID-19 outbreak in South Korea: A nationwide population based, longitudinal cohort study"
I suggest to revise the title to "Aspects of psychiatric comorbidities in breast cancer patients due to COVID-19 outbreak in South Korea: A single-center longitudinal cohort study" or something like that to avoid the "nationwide population". Otherwise, you should clear state that there is only one such center in Korea.
Author Response
Comments of Reviewer: Dear authors, according to the text, "For this retrospective, single-center study, patients data were collected from January 1, 2008 to June 30, 2021, at the Ajou University Hospital" and thus it is not correct to title the manuscript as the "Aspects of psychiatric comorbidities in breast cancer patients in tertiary hospitals due to COVID-19 outbreak in South Korea: A nationwide population based, longitudinal cohort study"
I suggest to revise the title to "Aspects of psychiatric comorbidities in breast cancer patients due to COVID-19 outbreak in South Korea: A single-center longitudinal cohort study" or something like that to avoid the "nationwide population". Otherwise, you should clear state that there is only one such center in Korea.
Response=> Your opinion is definitely correct and modified title.